



# Impacts of biomass burning and photochemical processing on the light absorption of brown carbon in the southeastern Tibetan Plateau

Jie Tian[1,2], Qiyuan Wang[1,2], Yongyong Ma[3], Jin Wang[1], Yongming Han[1,2], and Junji Cao[1,4]

[1]Key Laboratory of Aerosol Chemistry and Physics, State Key Laboratory of Loess and Quaternary Geology, Institute of Earth
Environment, Chinese Academy of Sciences, Xi'an 710061, China
[2]CAS Center for Excellence in Quaternary Science and Global Change, Xi'an 710061, China
[3]Meteorological Institute of Shaanxi Province, Xi'an 710015, China
[4]Institute of Atmospheric Physics, Chinese Academy of Sciences, Beijing 100029, China

*Correspondence*: Qiyuan Wang (wangqy@ieecas.cn) and Junji Cao (jjcao@mail.iap.ac.cn)

**Abstract.** Brown carbon (BrC) in the atmosphere can greatly influence aerosol's radiative forcing over the Tibetan
Plateau (TP), because it has the non-negligible capacity of light absorption as well as black carbon (BC); however, our
understanding of optical properties, sources, atmospheric processes of BrC in this region remains limited. In this study,
a multiple-wavelength Aethalometer coupled with a quadrupole aerosol chemical speciation monitor were deployed to
investigate the highly time resolved BrC in the submicron aerosol in the southeastern edge of the TP during the pre-
monsoon season. The result showed that BrC had the substantial contributions (20.0–40.2 %) to the light absorption of
submicron aerosol from 370 to 660 nm. Organic aerosol (OA), an alternative to BrC, was split into a biomass burning
OA (BBOA) with aging process and a photochemical-oxidation processed oxygenated OA (po-OOA) by a hybrid
environmental receptor model analysis. Combined with light absorption coefficient of BrC ($b_{abs-BrC}$), the source-specific
mass absorption cross section of BBOA (0.50−1.75 $m^2$ $g^{-1}$) and po-OOA (0.38−2.15 $m^2$ $g^{-1}$) at 370−660 nm were
retrieved. On average, $b_{abs-BrC}$ from po-OOA (1.1−6.3 $Mm^{-1}$) was higher than that from BBOA (0.7−2.3 $Mm^{-1}$) at all
wavelengths. The concentration weighted trajectory analysis showed that the most important potential source regions
for $b_{abs-BrC}$ values from BBOA and po-OOA were located in the northern Myanmar and along the China-Myanmar border,
indicating the cross-border transport of BrC from Southeast Asia. A "simple forcing efficiency" evaluation further
illustrated the importance of BrC radiative effect with the high fractional radiative forcing by two OAs relative to BC
(50.6 ± 18.7 %). This study highlighted a significant influence of BrC from biomass burning emissions and secondary
formation on climate change over the TP region during the pre-monsoon season.





# 1 Introduction

Carbonaceous aerosols, a major component of atmospheric particles, play an important role in the global climate by
directly absorbing and scattering solar and terrestrial radiation (Bellouin et al., 2013; IPCC, 2013; Yao et al., 2017). In
the past, black carbon (BC) was often considered to be the only light-absorbing carbonaceous aerosol, and organic
aerosol (OA) was thought to purely scatter light (Bond and Bergstrom, 2006; Koch et al., 2007). However, a fraction of
OA has been found to absorb radiation efficiently in near-ultraviolet (UV) and visible spectral ranges with a strong
wavelength dependence (Kirchstetter et al., 2004). This light-absorbing OA, collectively known as brown carbon (BrC)
(Andreae and Gelencsér, 2006), is receiving increasing attention due to its non-negligible radiative effect. Feng et al.
(2013) and Lin et al. (2014) have reported that the radiative forcing (RF) caused by BrC absorption on a global scale can
be up to + 0.25 and + 0.57 W m$^{-2}$, respectively. Zhang et al. (2020a) estimated that globally BrC contributed more than
25% of BC RF. In particular, the atmospheric heating of BrC is greater than that of BC in the tropical mid and upper
troposphere. Zhang et al. (2017) also suggested that the clear-sky RFs of high- and low-altitude BrC were 0.35 $\pm$ 0.16
and 0.65 $\pm$ 0.34 W m$^{-2}$, corresponding to 34% and 24% of carbonaceous aerosol warming effect at the tropopause,
respectively. The inclusion of BrC in climate models can reduce uncertainties in the global or regional RF assessment
of aerosols. Therefore, a comprehensive understanding on light absorption properties of BrC is required.

In the atmosphere, primary BrC is mainly emitted from biomass burning and fossil fuel combustion (Olson et al., 2015;
Washenfelder et al., 2015; Xie et al., 2017), and secondary BrC is commonly formed from photochemical-oxidation and
aqueous reactions of biogenic or anthropogenic precursors (Hecobian et al., 2010; Nakayama et al., 2013). The complex
sources and formation mechanisms of BrC lead to the spatial-temporal variations in its light absorption properties.
Accurately quantifying the source-specific absorption capacity (i.e., mass absorption cross section (MAC)) of BrC is
essential for modelling BrC climate effect. However, direct source apportionment of BrC is impossible with current
analytical method, since BrC constituents responsible for light absorption remain relatively unknown (Laskin et al.,
2015). Recent studies have usually used OA as an alternative to BrC for two major reasons: (1) the definition of OA
contains all BrC constituents, and (2) OA from either online monitoring or filter extraction can be apportioned to a few
major primary and secondary sources with the development of mass spectrometry. This allows for establishing the
relationship between primary and secondary OA types and BrC absorption, which provides better quantification of the
impact of BrC from different sources and formation mechanisms on regional and global climates (Kaskaoutis et al., 2021;
Moschos et al., 2018; Qin et al., 2018; Wang et al., 2021).

The Tibetan Plateau (TP), often referred to as the "Third Pole", is the largest and highest mountain region in the world
and contains the most abundant ice outside of the polar regions (Yao et al., 2012). It has a huge impact on the large-scale
atmospheric circulation and the hydrological cycle, which is the most sensitive and visible indicator of climate change





in the entire Asian continent (Chen and Bordoni, 2014; Immerzeel et al., 2010). In the recent decades, there has been the growing evidence of increased surface temperature in the Himalayas and the TP regions, accompanied by the accelerated glacier melt and retreat (Kehrwald et al., 2008; Liu and Chen, 2000; Wang et al., 2008). This rapid warming was firstly attributed to greenhouse gas warming; however, light-absorbing aerosols were found to be another major warming agent (Lau et al., 2010; Ramanathan et al., 2007). Previous studies paid a large amount of attention on BC due to its vital climatic implication in the TP. The sources of BC varied significantly with the receptor location and the season (Kopacz et al., 2011; Tan et al., 2021; Zhang et al., 2015). For example, Zhang et al. (2015) reported that biomass burning from South Asia has the largest impact on BC in the central plateau, while fossil fuel combustion contributed the most to BC burden in the northeast plateau in all seasons and southeast plateau in the summer. The direct RF of BC at the top of the atmosphere (+ 1.6−3.5 W m$^{-2}$) induced atmospheric heating rates of 0.13−0.35 K day$^{-1}$ in the Himalayas and the TP regions (Liu et al., 2021; Panicker et al., 2020); meanwhile, BC deposited on the snow-covered areas can increase 1.0 °C of the surface temperature over the TP by reducing the snow albedo (Qian et al., 2011). It is clearly that there are primary OA emissions along with BC emitted from biomass burning and fossil fuel combustion, and secondary OA formation has been found in the TP (Xu et al., 2018; Zhang et al., 2019). However, the link between light absorption properties and sources of OA is less understood so far, which would lead to uncertainties in evaluating aerosol radiative effect of TP.

In this study, real-time measurements of both light absorption properties and chemical characteristics of submicron aerosol were conducted in the southeastern margin of the TP during the pre-monsoon season. The main objectives were to (1) characterize the light absorption properties of BrC, (2) quantify the source-specific MAC and absorption of BrC, and (3) evaluate the importance of BrC radiative effect from different sources. This study provides insights into light absorption properties of BrC, which is necessary for understanding the role of BrC in climate warming and revealing impacts of sources and atmospheric processes in the TP and surrounding areas.

## 2 Methodology

### 2.1 Sampling site and period

Submicron aerosol online measurements of optical and chemical properties were performed at the Lijiang Astronomical Station, Chinese Academy of Sciences, Gaomeigu County, Yunnan Province (26 °41' N, 100 °1' E; 3260 m a.s.l.) (Fig. 1). Continuous hourly O$_3$ and relative humidity (RH) were measured with the use of an ozone analyzer (EC9810, Ecotech Pty Ltd, Australia) and an automatic weather station (MAWS201, Vaisala, Helsinki, Finland). The monitoring station is situated in the southeastern edge of the TP, a natural channel for the transport of air pollutants from Southeast Asia to the TP (Tan et al., 2021). All instruments were placed on the rooftop of an office building (~10 m above the ground), without any strong anthropogenic emission sources nearby. More detailed description about the sampling site can be





found in Wang et al. (2019) and Liu et al. (2021). The sampling period lasted from 8:00 local stand time (LST, all time
references that follow are given in LST) on 14 to 23:00 on 31 March, 2018, corresponding to the pre-monsoon season
in the TP.

**2.2 Submicron aerosol measurements**

A newly developed Aethalometer (Model AE33, Magee Scientific, Berkeley, CA, USA) was used to measure aerosol
light absorption coefficient ($b_{abs}$) at multiple wavelengths (i.e., 370, 470, 520, 590, 660, and 880 nm) with a 1 min time
resolution. Briefly, the ambient air sampled at a flow rate of 5 L min$^{-1}$ was firstly selected by a PM$_1$ (particulate matter
with an aerodynamic diameter $\leq$ 1.0 μm) cyclone separator (BGI SCC 1.197, Mesa Labs, USA) to collect submicron
aerosol on the filter. Light at different wavelengths emitted from diodes irradiated two parallel filter spots with deposition
rates of 3.85 and 1.15 L min$^{-1}$, respectively. Thereafter, the two light attenuations measured by optical detectors was
used to calculate $b_{abs}$ through a real-time loading effect compensation algorithm. This "dual-spot" technique for the
Model AE33 can eliminate the nonlinearity effect caused by increasing amount of aerosol deposit on the filter
(Weingartner et al., 2003). Additionally, the Model AE33 automatically used a factor of 2.14 to compensate the scattering
effect of quartz filter. Detailed operating principles of the Model AE33 can be found in Drinovec et al. (2015).

OA in the non-refractory PM$_1$ was measured using a quadrupole aerosol chemical speciation monitor (Q-ACSM,
Aerodyne Research Inc., Billerica, Massachusetts, USA) with a 30 min time resolution. The aerodynamic lens coupled
with a 100 μm diameter critical aperture in the Q-ACSM created a beam of focused submicron aerosols (40−1000 nm
aerodynamic diameter), which was vaporized at ~600 °C, ionized by a 70 eV electron impact, and subsequently
characterized with a mass spectrometer. The details of the instrument have been described elsewhere (Ng et al., 2011b).
The measured Q-ACSM data was processed by the ACSM local tool version 1.5.3.5 compiled with Igor Pro 6.37
(Wavemetrics, Inc., Lake Oswego, OR, USA) to determine the mass concentration and ion-speciated mass spectra of
OA. Four our study, the default collection efficiency (0.45) and relative ionization efficiency (1.4) were used to obtain
OA mass concentration (Middlebrook et al., 2012). The mass concentration and error matrices of organic fragments from
mass-to-charge ($m/z$) 12 to 120 were initialized following the method of Allan et al. (2003).

**2.3 Data analysis**

**2.3.1 Separation of BrC and BC absorption**

The extrapolation method based on Absorption Ångström exponent (AAE) is widely used to project the absorption at
the longer wavelength to the shorter wavelength of the spectrum (Olson et al., 2015; Pokhrel et al., 2017). With an
assumption of $b_{abs}$ at 880 nm ($b_{abs}$ (880 nm)) solely from BC (Kirchstetter et al., 2004), the light absorption coefficient
of BC ($b_{abs\text{-}BC}$) at wavelengths (λ) of 370, 470, 520, 590, and 660 nm can be obtained using the following formula:





$$b_{\text{abs-BC}}\,(\lambda) = b_{\text{abs}}\,(880\ \text{nm}) \times \left(\frac{880}{\lambda}\right)^{\text{AAE}_{\text{BC}}} \tag{1}$$

Here, $b_{\text{abs}}$ and $b_{\text{abs-BC}}$ are given in inverse megameters (Mm$^{-1}$); AAE$_{\text{BC}}$ is assumed to be 1.1 ±0.3, which represents the likely range of AAE for BC externally and internally mixed with non-absorbing material (Lack and Langridge, 2013). Then, BrC absorption is derived by subtracting BC absorption from the total submicron aerosol absorption via:

$$b_{\text{abs-BrC}}\,(\lambda) = b_{\text{abs}}\,(\lambda) - b_{\text{abs-BC}}\,(\lambda) \tag{2}$$

Here, $b_{\text{abs-BrC}}$ is the light absorption coefficient of BrC (Mm$^{-1}$).

**2.3.2 Hybrid environmental receptor model (HERM) analysis**

HERM analysis was performed to retrieve potential sources of OA, using mass concentration and error matrices of organic fragments measured by the Q-ACSM. The principle of HERM has been described elsewhere (Chen and Cao, 2018). Briefly, the HERM is a bilinear receptor model, which decomposes measured organic fragments matrix (X) at the receptor into matrices of the source contributions (G), source mass spectra (F), and the model residual (E):

$$X = G \times F + E \tag{3}$$

The HERM algorithm attempts to solve G and F by minimizing the object function Q, defined as:

$$Q = \sum_{i=1}^{I} \sum_{j=1}^{J} \frac{\left(x_{ij} - \sum_{k=1}^{K} g_{ik} f_{kj}\right)^2}{\sigma_{x_{ij}}^2 + \sum_{k=1}^{K}\left(g_{ik}^2 \sigma_{f_{kj}}^2 + \delta_{jk}\sigma_{x_{ij}}^2\right)} \tag{4}$$

Here, I, J, and K are the number of samples, $m/z$ variables, and sources, respectively; $x_{ij}$ is the measured concentration of the $j$th $m/z$ in the $i$th sample; $g_{ik}$ is the contribution of the $j$th source in the $i$th sample; $f_{kj}$ is the $j$th $m/z$ fraction of the

total organic fragments in the $k$th source (so-called mass spectrum); $\sigma_{x_{ij}}$ and $\sigma_{f_{kj}}$ represent the error in measured $m/z$ concentration and the variability in constrained mass spectrum, respectively; $\delta_{jk}$ is set to 0 or 1 depending on whether the $j$th $m/z$ in the $k$th mass spectrum is constrained or unconstrained, respectively.

Before HERM analysis, $m/z$ from 12 to 120 with signal-to-noise between 0.2 and 2 and $m/z$ 44 were down-weighted by increasing their errors by a factor of 2 (Ulbrich et al., 2009). HERM solutions from two to five factors with unconstrained

mass spectrum were investigated to explore potential sources. The two-factor solution were chosen as the optimal solution, while greater number of factors (3−5) solutions existed many non-physical meaning factors dominated by individual $m/z$ and do not further split new sources. Bootstrap (BS) method was adopted for two-factor solution (Brown et al., 2015). In 50 times BS, no mass spectrum was unmapped (r < 0.6) indicating the two-factor solution was robust. Therefore, a biomass burning OA (BBOA) and a photochemical-oxidation processed oxygenated OA (po-OOA) were



finally identified. More detailed description of mass spectra, time series, and correlations with tracers of these two OA factors can be found in Sect. 3.2.

### 2.3.3 Calculation of optical parameters

MAC, expressed by normalized absorption cross sections to the mass of particles, is commonly used to describe the light absorption capacity of aerosols (Bond and Bergstrom, 2006). The MAC of OA component in this study was resolved by

the multiple linear regression (MLR) model combined with $b_{\text{abs-BrC}}$ and OA source apportionment results. The amount of $b_{\text{abs-BrC}}$ at different wavelengths can be estimated as follows:

$$b_{\text{abs-BrC}}(\lambda) = a_1(\lambda) \times [\text{BBOA}] + a_2(\lambda) \times [\text{po-OOA}] \tag{5}$$

Here, $a_1$ and $a_2$ denote the MAC of BBOA ($\text{MAC}_{\text{BBOA}}$) and po-OOA ($\text{MAC}_{\text{po-OOA}}$), respectively, in square meters per gram ($\text{m}^2\ \text{g}^{-1}$); [BBOA] and [po-OOA] are the mass concentration of BBOA and po-OOA, respectively, in micrograms

per cubic meter ($\mu\text{g}\ \text{m}^{-3}$). The MAC of BC ($\text{MAC}_{\text{BC}}$) was directly calculated with $b_{\text{abs-BC}}$ divided by the mass concentration of BC [BC]:

$$\text{MAC}_{\text{BC}}(\lambda) = \frac{b_{\text{abs-BC}}(\lambda)}{[\text{BC}]} \tag{6}$$

Here, [BC] is obtained by dividing $b_{\text{abs}}$ (880 nm) by the default MAC (880 nm) used in the Model AE33 ($\mu\text{g}\ \text{m}^{-3}$) (Drinovec et al., 2015).

AAE describes the spectral dependence of light absorption by aerosols, and it often reflect the composition of light-absorbing components. Generally, the greater proportion of BrC relative to BC indicates the larger AAE (Lack and Cappa, 2010). Equations (7) and (8) show the calculations of AAE using a power law function with $b_{\text{abs}}$ and MAC, respectively:

$$b_{\text{abs}}(\lambda) = k_1 \times \lambda^{-\text{AAE}} \tag{7}$$

$$\text{MAC}(\lambda) = k_2 \times \lambda^{-\text{AAE}} \tag{8}$$

Here, $k_1$ and $k_2$ are constants independent of wavelength.

### 2.3.4 Statistical metrics

The uncentered correlation coefficient (UC) is a qualitative metric to characterize the similarity between mass spectra of sources, which is defined as follows (Ulbrich et al., 2009):

$$\text{UC} = \frac{x \cdot y}{\|x\|\ \|y\|} \tag{9}$$

Here, $x$ and $y$ represent a pair of mass spectra as vectors.





The index of agreement (IOA) is used as an indicator to evaluate the performance of the simulated data from MLR model against the measured data (Willmott, 1981). The IOA varies between 0 (no agreement) and 1 (perfect agreement), and can be expressed as:

$$IOA = 1 - \frac{\sum_{i=1}^{N}(S_i - M_i)^2}{\sum_{i=1}^{N}(|S_i - M_{ave}| + |M_i - M_{ave}|)^2} \tag{10}$$

Here, N is the total number of the simulated data; $S_i$ and $M_i$ are the simulated and measured $b_{abs\text{-}BrC}$, respectively; and $M_{ave}$ is the average measured $b_{abs\text{-}BrC}$.

## 2.4 Trajectory-related analysis

A geographic information system based software TrajStat was utilized to investigate the influences of regional transport on BrC absorption at Gaomeigu from 14 to 31 March, 2018 (Wang et al., 2009). The trajectories were calculated with the Hybrid Single-Particle Lagrangian Integrated Trajectory (HYSPLT) model developed by the National Oceanic and Atmospheric Administration (Draxler and Hess, 1988). In this study, the model was set to run twenty-four times per day at starting times of 0:00−23:00 with 1 h step. 72-h backward trajectories at the height of 500 m above the ground level at Gaomeigu during the sampling period were produced based on the gridded meteorological data from Global Data Assimilation System (ftp://arlftp.arlhq.noaa.gov/pub/archives/gdas1, last access: 1 June, 2022).

The concentration weighted trajectory (CWT) method was further used to identify the potential source regions that likely affected the BrC absorption at Gaomeigu (Hsu et al., 2003). The geographic zone covered by the total number of backward trajectories (K) was divided into I × J grid cells with the resolution of 0.5 ° × 0.5 °. The CWT value of each grid can be defined as follows:

$$b_{abs\text{-}BrC\text{-}ij} = \frac{\sum_{k=1}^{K} b_{abs\text{-}BrC\text{-}k} \tau_{ijk}}{\sum_{k=1}^{K} \tau_{ijk}} W_{ij} \tag{11}$$

Here, $b_{abs\text{-}BrC\text{-}ij}$ is the average weighted light absorption coefficient of BrC in the $ij$th cell; $b_{abs\text{-}BrC\text{-}k}$ is the light absorption coefficient of BrC observed on the arrival of trajectory $k$; and $\tau_{ijk}$ is the time spent in the $ij$th cell by trajectory $k$. The weighting function of $W_{ij}$ was applied to reduce the effect of the small number of back-trajectory segment endpoints that fall into the grid cell (Wang et al., 2006):

$$W_{ij} = \begin{cases} 1.00 & 135 < n_{ij} \\ 0.70 & 45 < n_{ij} \leq 135 \\ 0.42 & 15 < n_{ij} \leq 45 \\ 0.17 & n_{ij} \leq 15 \end{cases} \tag{12}$$

Here, $n_{ij}$ is the total number of endpoints in the $ij$th cell. In this study, the total number of endpoints located in 72 cells of the geographic zone is 3268, so that the average number of endpoints in all cells is about 45.



## 2.5 Radiative effect calculation

The concept "simple forcing efficiency" (SFE) introduced by Bond and Bergstrom (2006) is a useful way to evaluate
the radiative effect of atmospheric aerosols. Without consideration of aerosol scattering, a variant of wavelength-dependent SFE is given as follows:

$$\text{SFE}_i\,(\lambda) = \frac{S_0\,(\lambda)}{4} \times \tau_{atm}^2 \times (1 \; - \; F_c) \times [4a_s \times \text{MAC}_i\,(\lambda)] \tag{13}$$

Here, the subscript $i$ represents BBOA, po-OOA, or BC; $\lambda$ denotes the wavelength from 370 to 660 nm with 1 nm step;
SFE is given in watts per gram (W g$^{-1}$), which represents the positive energy added to the Earth atmosphere system by a
given mass of light-absorbing particles in the atmosphere; $S_0$ is the solar irradiance based on the ASTM G173-03
reference spectra in watts per square meters (W m$^{-2}$); $\tau_{atm}$, $F_c$, and $a_s$ are the atmospheric transmission (0.79), the cloud
fraction (0.6), and the surface albedo (0.9), respectively, which are constants from the global average calculations; MAC
with a 1 nm resolution is extrapolated using Eq. (8). And then, the fraction of solar radiation absorbed by OA component
relative to BC ($f_{\text{OA/BC}}$) can be calculated as:


$$f_{\text{OA/BC}} = \frac{\sum_{\lambda=370}^{660} \text{SFE}_{\text{OA}}\,(\lambda) \times C_{\text{OA}}}{\sum_{\lambda=370}^{660} \text{SFE}_{\text{BC}}\,(\lambda) \times C_{\text{BC}}} \tag{14}$$

Here, the integrated SFE is the sum of the SFE from 370 to 660 nm; $C_{\text{OA}}$ and $C_{\text{BC}}$ are the average mass concentrations
of OA component and BC during the sampling period.

## 3 Results and discussion

### 3.1 Overview of BrC absorption

The temporal variation in submicron aerosol $b_{\text{abs}}$ at wavelengths from 370 to 880 nm as well as the OA mass
concentration during the entire campaign at Gaomeigu are depicted in Fig. S1 in the Supplement. The hourly $b_{\text{abs}}$ values
varied 18−41 folds from 14 to 31 March 2018, reflecting that atmospheric environment at Gaomeigu is influenced by
dynamic changes in emission sources and meteorological condition. Particularly, the larger variations in $b_{\text{abs}}$ values at
370−660 nm, compared with that at 880 nm, highlighted the impact of non-BC light-absorbing materials. As shown in
Table 1, the average $b_{\text{abs}}$ values were 33.1 $\pm$ 24.4 Mm$^{-1}$ (arithmetic mean $\pm$ standard deviation) at 370 nm, 26.7 $\pm$ 19.7
Mm$^{-1}$ at 470 nm, 20.3 $\pm$ 13.9 Mm$^{-1}$ at 520 nm, 18.2 $\pm$ 12.5 Mm$^{-1}$ at 590 nm, 13.7 $\pm$ 9.0 Mm$^{-1}$ at 660 nm, and 8.0 $\pm$ 4.9
Mm$^{-1}$ at 880 nm. The $b_{\text{abs}}$ values obtained in this study were comparable with those reported previously at the sampling
sites of the TP, where the major anthropogenic sources (i.e., industry, fossil fuel combustion, etc.) are limited locally
(Niu et al., 2018; Zhao et al., 2019; Zhu et al., 2021). Frequency histograms of hourly AAE values showed a typical





normal distribution, with an average AAE value of 1.62 ±0.28 (Fig. S2). Over 72 % of AAE values were higher than 1.4 (Upper limit of $AAE_{BC}$), implying the presence of both BrC and BC absorption in the submicron aerosol at Gaomeigu.

Based on Eqs. (1) and (2), $b_{abs-BrC}$ and $b_{abs-BC}$ were separated from the total absorption using the $AAE_{BC}$ = 1.1. The average $b_{abs-BrC}$ values during the campaign were 12.3 ±13.8, 10.7 ±11.4, 6.0 ±6.0, 5.8 ±5.7, and 2.7 ±2.6 Mm$^{-1}$ at 370, 470, 520, 590, and 660 nm, respectively (Table 1). Figure 2 shows fractions of light absorption at specific

wavelengths by BrC and BC in the submicron aerosol. BrC contributed substantially to $b_{abs}$, which accounted for 20.0–40.2 % from 370 to 660 nm. The average contributions of $b_{abs-BrC}$ to $b_{abs}$ in the near-UV and blue spectral ranges (300–500 nm) were higher than those in other visible ranges (520–880 nm), indicating that BrC was a considerable absorbing material at short wavelengths in the atmosphere. It should be noted that the assumption for $AAE_{BC}$ would lead to the biases in the BrC absorption calculation (Lack and Langridge, 2013). The uncertainties of $AAE_{BC}$ (±0.3) in this study

caused variations of 14.3−16.6 %, 10.2−12.2 %, 10.3−11.5 %, 7.7−8.6 %, and 6.6−7.1 % in the estimation of BrC absorption contributions at 370, 470, 520, 590, and 660 nm, respectively (Fig. 2).

With respect to the relationship between BrC absorption and OA components, $b_{abs-BrC}$ values at 370–660 nm were significantly correlated with OA mass concentrations (r = 0.64−0.70, $p < 0.01$) (Fig. S3), confirming a strong link between BrC-chromophores and OA in the southeastern margin of TP (Lack et al., 2013; Laskin et al., 2015).

**3.2 OA source apportionment**

HERM analysis identified two distinct OA sources, consisting of BBOA and po-OOA. Each of OA sources had unique characteristics on mass spectrum, temporary variation, and atmospheric processes. The detailed source apportionment results of OA are shown in Fig. 3.

The mass spectrum of BBOA resembled that of BBOA obtained in previous studies (UC = 0.80−0.87) (Crippa et al.,

2013; Ng et al., 2011a; Wang et al., 2017). It was characterized by a prominent peak of $m/z$ 60, and a strong positive correlation was found between BBOA and $m/z$ 60 concentrations (r = 0.72, $p < 0.01$) (Figs. 3a and b). We have known that $m/z$ 60 is a typical molecular fragment of levoglucosan, mannosan, and galactosan, which are good organic tracers of biomass burning (Kalogridis et al., 2018; Kim et al., 2017; Reyes-Villegas et al., 2018). The fraction of $m/z$ 60 in BBOA mass spectrum ($f_{60}$, 0.9 %) was higher than 0.3% (background level in absence of biomass burning), suggesting

the impact of biomass burning at Gaomeigu (Cubison et al., 2011). Scatterplots of $f_{44}$ vs. $f_{60}$ was used to analyze aging degree of BBOA in the atmosphere (Fig. 3c). The $f_{60}$ usually decreases from fresh to aged biomass burning emissions because of degradation and oxidation reactions during the atmospheric aging, while the $f_{44}$ increases (Paglione et al., 2020). The $f_{60}$ and $f_{44}$ of BBOA resolved in this study (0.9 % and 6.3 %, respectively) indicates BBOA was partly aged,





possibly caused by the long-distance regional transport. This is further demonstrated in our trajectory-related analysis
(Sect. 3.3).

Another OA source was featured by the high correlation with $m/z$ 44 (r = 0.97, $p < 0.01$), as well as previously reported
OOA (Tobler et al., 2021; Xu et al., 2018; Zhang et al., 2019). The mass spectrum of po-OOA had a high $f_{44}$ (27.8 %)
and a low $f_{43}$ (0.8 %), that was likely related to extensive secondary processes occurring during transport (Wang et al.,
2017; Xu et al., 2017). Figure 3d shows that both po-OOA mass concentration and its fraction in OA increased with
increasing $O_3$ ($R^2 = 0.79-0.87$), however, neither of them correlated with RH (Fig. S4). These results supported that
photochemical-oxidation process appeared to affect most formation of po-OOA. Moreover, the temporal variation of
mass concentration in po-OOA significantly correlated with that in BBOA (r = 0.63, $p < 0.01$), indicating that a portion
of po-OOA could be derived from oxidation of volatile organic precursor from biomass burning (Bruns et al., 2016;
Posner et al., 2018).

**3.3 Source-dependent characteristics of BrC absorption**

To further quantify the contributions of OA sources to BrC light absorption, a MLR model was applied to retrieve MAC
values from BBOA and po-OOA. As shown in Fig. 4a, the wavelength dependences of $MAC_{BBOA}$ and $MAC_{po-OOA}$ were
generally described by the power-law relationship ($R^2 = 0.77-0.87$), and the AAE of BrC values for BBOA and po-OOA
were greater than 2.0 (Kirchstetter et al., 2004). The $MAC_{BBOA}$ was $1.75 \pm 0.48$ m$^2$ g$^{-1}$ at 370 nm, and dropped to $0.50 \pm$
$0.08$ m$^2$ g$^{-1}$ at 660 nm. Taking the near-UV wavelength as the representative for discussion, the $MAC_{BBOA}$ obtained in
this study was within that range from biomass burning (0.9−7.7 m$^2$ g$^{-1}$) reported by laboratory experiments and field
measurements studies (Kaskaoutis et al., 2021; Kumar et al., 2018; Lack et al., 2012; Qin et al., 2018; Wang et al., 2020;
Washenfelder et al., 2015). The differences in light absorption capacity of OA from biomass burning may be partly
associated with biomass types and combustion efficiencies (Budisulistiorini et al., 2017; Martinsson et al., 2015; Tian et
al., 2019). In addition, the photobleaching effect of aerosol at different aging degree can also lead to the variation in
$MAC_{BBOA}$ (Sumlin et al., 2017; Zhong and Jang, 2014). The $MAC_{po-OOA}$ was $2.15 \pm 0.24$ m$^2$ g$^{-1}$ at 370 nm, $1.69 \pm 0.19$
m$^2$ g$^{-1}$ at 470 nm, $0.90 \pm 0.09$ m$^2$ g$^{-1}$ at 520 nm, $0.85 \pm 0.09$ m$^2$ g$^{-1}$ at 590 nm, and $0.38 \pm 0.04$ m$^2$ g$^{-1}$ at 660 nm. Compared
with BBOA, po-OOA absorbed light more efficiently in the near-UV, consisting with previous findings for
photochemical production of BrC from biomass burning (Kumar et al., 2018; Moschos et al., 2018).

The MLR model reasonably reconstructed the temporal variation in the measured $b_{abs-BrC}$ values, with IOAs ranged from
0.73 to 0.79 for different wavelengths. As shown in Fig. 4b, the average $b_{abs-BrC}$ values from BBOA were $2.3 \pm 0.3$, $2.2$
$\pm 0.3$, $1.3 \pm 0.4$, $1.2 \pm 0.4$, and $0.7 \pm 0.4$ Mm$^{-1}$ at 370, 470, 520, 590, and 660 nm, respectively. The $b_{abs-BrC}$ from BBOA
had considerable contributions (29.5−40.2 %) to the total reconstructed light absorption coefficient of BrC ($rb_{abs-BrC}$) at





370−660 nm, indicating that biomass burning is an important primary source of BrC absorption at Gaomeigu. This was possibly related to transport of pollutants emitted from South and Southeast Asia during the pre-monsoon season, where biomass burning activities are intensive (Zhang et al., 2020b; Zhang et al., 2015). The po-OOA produced larger $b_{abs-BrC}$ values (6.3 ±0.7, 4.9 ±0.7, 2.6 ±0.6, 2.5 ±0.6, and 1.1 ±0.6 Mm$^{-1}$ at 370, 470, 520, 590, and 660 nm, respectively), suggesting the critical role of photochemical-oxidation processes in the BrC absorption at Gaomeigu. Four periods are marked in Fig. S1, characterized by the inclusion of obvious rising stages for both OA concentrations and $b_{abs}$ values. The $b_{abs-BrC}$ at 370 nm from BBOA and po-OOA contributed a comparable faction (42.1 % vs. 57.9 %) to r$b_{abs-BrC}$ during the period I, while the contribution of $b_{abs-BrC}$ from po-OOA increased significantly during other periods (72.3−81.4 %) (Fig. S5). The rapid increases in $b_{abs-BrC}$ from po-OOA were likely caused by photochemical-oxidation processes that were favored by relatively high O$_3$ condition (75−84 ppb) during the periods II−IV; for comparison, the O$_3$ mixing ratio during period I was 52 ppb. We noted that the largest r$b_{abs-BrC}$ occurred in period II when both primary source emissions and secondary formation were strong. These results further highlighted the importance of biomass burning and photochemical-oxidation on light absorption of BrC at Gaomeigu.

The air-mass trajectory and CWT analyses were used to identify whether local emission or regional transported air pollution was the major source of $b_{abs-BrC}$ from OA components at Gaomeigu. Figure 5a shows the 72-h backward trajectories of the receptor site during the sampling period, and all those were originated from Myanmar. The percentage of the trajectories with high OA concentration (> 5.3 μg m$^{-3}$, the median value of hourly OA concentration) exceed 50%. From the CWT maps of $b_{abs-BrC}$ at 370 nm, the spatial distributions of potential source for $b_{abs-BrC}$ from BBOA and po-OOA were similar (Figs. 5b and c). The source regions with the highest CWT values were located in the northern Myanmar and along the China-Myanmar border, while the CWT values in the areas surrounding Gaomeigu were relatively low. This indicates that large $b_{abs-BrC}$ loadings at Gaomeigu were more likely resulted in strong cross-border transport of BrC from biomass burning and secondary formation than local emission during the pre-monsoon season.

### 3.4 Radiative effect of BrC

As described in Sect. 2.5, a simple model was used to provide a first-order estimate of the radiative effect of light-absorbing particles. Figure 6a shows SFEs of BBOA, po-OOA, and BC at wavelengths from 370 to 660 nm. The SFE peaks of BBOA, po-OOA, and BC were observed at the boundary between the UV and blue spectra (i.e., ~450 nm), that was mainly caused by the high MAC and strong solar irradiance at specific wavelength. The integrated SFEs of BBOA and po-OOA over the entire solar spectra (370−660 nm) in this study were 17.4 and 17.0 W g$^{-1}$, respectively, comparable to that of primary OA (21 W g$^{-1}$, 300−1000 nm) reported in Lu et al. (2015). BC had a much higher integrated SFE (226.6 W g$^{-1}$) compared with OAs. This is consistent with the widely acknowledged view that BC is the strongest and most important light-absorbing particle in the atmosphere (Bahadur et al., 2012; Bond et al., 2013). As the concentration





of OA in the atmosphere is generally greater than that of BC, the importance of BrC radiative effect was further evaluated

by calculating the fraction of solar radiation absorbed by OA relative to BC. The fractional radiative forcing by two OAs

relative to BC was as high as $50.6 \pm 18.7$ %, in which the relative radiative forcing of po-OOA to BC ($32.9 \pm 18.0$ %)

was almost twice that of BBOA to BC ($17.7 \pm 6.6$ %) (Fig. 6b). These results suggested that the radiative effect of BrC,

especially secondary BrC, should be given more consideration in tackling climate change of the southeastern TP.

## 4 Conclusion

This study conducted an intensive real-time measurement at Gaomeigu in the southeastern margin of the TP during the

pre-monsoon season to investigate light absorption properties, sources, secondary formation, and radiative effect of BrC

in the submicron aerosol. Based on the assumption of $AAE_{BC} = 1$, the average $b_{abs-BrC}$ values were calculated as $12.3 \pm$

$13.8$, $10.7 \pm 11.4$, $6.0 \pm 6.0$, $5.8 \pm 5.7$, and $2.7 \pm 2.6$ Mm$^{-1}$ at 370, 470, 520, 590, and 660 nm, respectively, which

contributed 20.0–40.2 % of the total light absorption. OA was used as an alternative to BrC due to the significant

correlation (r = 0.64−0.70, $p < 0.01$) between its mass concentration and $b_{abs-BrC}$. The HERM analysis identified two OA

sources, including a BBOA and a po-OOA. BBOA was partly aged as evidenced by the $f_{60}$ (0.9 %) and $f_{44}$ (6.3 %) of

mass spectrum, while significant positive correlation between po-OOA and $O_3$ indicated that photochemical-oxidation

process was possibly the main pathway for the formation of po-OOA. A MLR model combined with $b_{abs-BrC}$ and OA

concentration was used to estimate the MAC of OA. The result showed that po-OOA absorbed light more efficiently in

the near-UV compared with BBOA. The $MAC_{BBOA}$ and $MAC_{po-OOA}$ was $1.75 \pm 0.48$ and $2.15 \pm 0.24$ m$^2$ g$^{-1}$ at 370 nm,

and dropped to $0.50 \pm 0.08$ and $0.38 \pm 0.04$ m$^2$ g$^{-1}$ at 660 nm, respectively. $b_{abs-BrC}$ from BBOA contributed 29.5−40.2 %

of the reconstructed $b_{abs-BrC}$ at 370−660 nm, while the rest was from po-OOA. All the 72-h backward trajectories of the

Gaomeigu site came from Myanmar. The spatial distributions of potential source regions for $b_{abs-BrC}$ showed the highest

CWT values for BBOA and po-OOA were both in the northern Myanmar and along the China-Myanmar border,

demonstrating that biomass burning emissions and secondary formation from the cross-border transport of Southeast

Asia were the major source of $b_{abs-BrC}$ at Gaomeigu. According to the integrated SFEs of BBOA, po-OOA, and BC over

the solar spectra (370−660 nm) (17.4, 17.0, and 226.6 W g$^{-1}$, respectively), the fractional radiative forcing by BBOA

($17.7 \pm 6.6$ %) and po-OOA ($32.9 \pm 18.0$ %) relative to BC were obtained, highlighting the importance of BrC radiative

effect. This study provides insights into light absorption properties of BrC and its potential impacts on climate change

over the TP and surrounding areas.



*Data availability.* Data used to support the findings in this study are archived at the Institute of Earth Environment, Chinese Academy of Sciences, and are publicly available at https://doi.org/10.5281/zenodo.7034650.

*Competing interests.* The authors declare that they have no conflict of interest.

*Author contributions.* QW, YH, and JC designed the campaign. JT and JW conducted field measurements. QW, JT, and YM made data analysis and interpretation. JT wrote the paper with contributions from all co-authors.

*Acknowledgments.* The authors are grateful to Weikang Ran, Yonggang Zhang, and other staff at the sampling sites for their assistance with field sampling.

*Financial support.* This research was supported the National Natural Science Foundation of China (grant no. 41877391),
the Second Tibetan Plateau Scientific Expedition and Research Program (STEP) (grant no. 2019QZKK0602), and the Youth Innovation Promotion Association of the Chinese Academy of Sciences (grant no. 2019402 and 2022416).

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





**Table 1.** Submicron aerosol light absorption coefficient ($b_{abs}$) contributed by BrC ($b_{abs\text{-}BrC}$) and BC ($b_{abs\text{-}BC}$) at Gaomeigu during the sampling period (March 14th to 31th, 2018).

| Parameter[*] ($Mm^{-1}$) | Wavelength | | | | | |
|---|---|---|---|---|---|---|
| | 370 nm | 470 nm | 520 nm | 590 nm | 660 nm | 880 nm |
| $b_{abs}$ | 33.1 ±24.4 (4.7−160.0)[**] | 26.7 ±19.7 (3.8−138.4) | 20.3 ±13.9 (2.6−93.0) | 18.2 ±12.5 (2.1−84.8) | 13.7 ±9.0 (1.7−56.9) | 8.0 ±4.9 (1.5−28.6) |
| $b_{abs\text{-}BrC}$ | 12.3 ±13.8 | 10.7 ±11.5 | 6.0 ±6.0 | 5.8 ±5.8 | 2.7 ±2.6 | 0.0 ±0.0 |
| $b_{abs\text{-}BC}$ | 20.8 ±12.8 | 16.0 ±9.8 | 14.3 ±8.8 | 12.4 ±7.7 | 11.0 ±6.8 | 8.0 ±4.9 |

[*]$b_{abs\text{-}BrC}$ and $b_{abs\text{-}BC}$ were calculated based on the $AAE_{BC}$ = 1.1.

[**]Variations of the measured hourly $b_{abs}$.





**Figure captions:**

**Figure 1.** Topography map of the Tibetan Plateau and the location of the sampling site at Gaomeigu.

**Figure 2.** Light absorption fractions at specific wavelengths contributed by BrC and BC under different absorption Ångström exponent of BC ($AAE_{BC}$) assumptions. The red, blue, and green lines were the dividing lines between BrC and BC light absorption fractions when $AAE_{BC} = 1.1$, 0.8, and 1.4, respectively. The grey filled area represents variations
in the BrC absorption fraction caused by the uncertainties of $AAE_{BC}$ ($\pm 0.3$).

**Figure 3.** (a) Mass spectra of BBOA and po-OOA. (b) Pearson correlations between mass concentrations of OA components and the tracer molecular fragments. (c) Scatterplots of $f_{44}$ vs. $f_{60}$ for BBOA resolved in this study and reported by previous literatures. (d) Variations of po-OOA mass concentration and its fraction in OA as a function of $O_3$. The data are grouped in $O_3$ bins (10 ppb increment).

**Figure 4.** (a) The mass absorption cross section of BBOA and po-OOA ($MAC_{BBOA}$ and $MAC_{po-OOA}$, respectively) at five wavelengths ($\lambda = 370$, 470, 520, 590, and 660 nm). The circle and shaded area represent the mean MAC values and the standard deviations, respectively. The dashed line is power-law fit. (b) Light absorption coefficient of BrC ($b_{abs-BrC}$) from BBOA and po-OOA and its fraction in the total reconstructed BrC absorption at different wavelengths.

**Figure 5.** (a) 72-h backward trajectories of Gaomeigu from 8:00 on 14 to 23:00 on 31 March, 2018. (b) and (c)
Concentration weighted trajectory (CWT) maps of $b_{abs-BrC}$ at 370 nm ($Mm^{-1}$) from BBOA and po-OOA, respectively.

**Figure 6.** (a) Simple forcing efficiency (SFE) of light-absorbing particles from 370 to 660 nm and the integrated SFE over the entire solar spectra (370−660 nm). (b) The fraction of solar radiation absorbed by OA components relative to BC. In each panel, the short line and square inside the boxes indicate the median and mean values, respectively. The lower and upper edges of the boxes denote the standard deviation. The vertical bars ("whiskers") show the 5[th] and 95[th]
percentiles. Scattered data points and normal distribution curve are also shown.



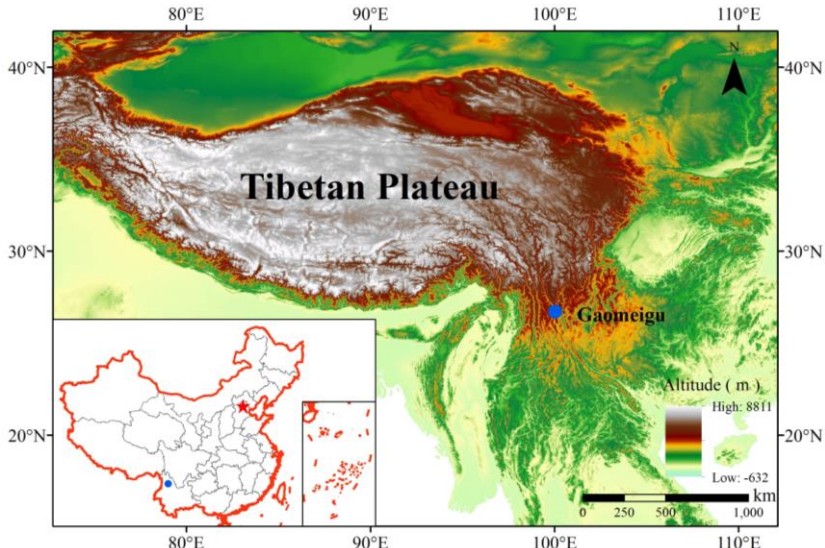

**Figure 1.** Topography map of the Tibetan Plateau and the location of the sampling site at Gaomeigu.





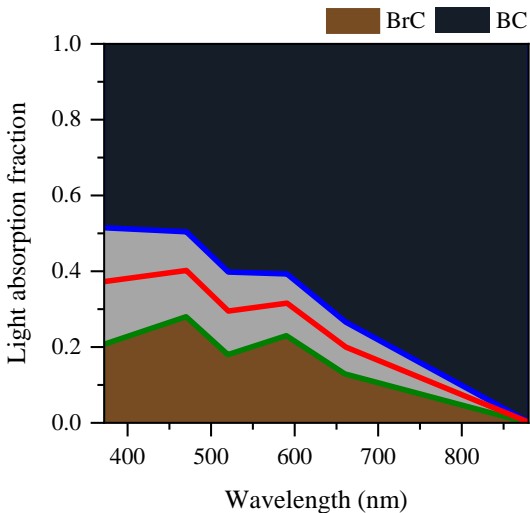

**Figure 2.** Light absorption fractions at specific wavelengths contributed by BrC and BC under different absorption
Ångström exponent of BC (AAE$_{BC}$) assumptions. The red, blue, and green lines were the dividing lines between BrC
and BC light absorption fractions when AAE$_{BC}$ = 1.1, 0.8, and 1.4, respectively. The grey filled area represents variations
in the BrC absorption fraction caused by the uncertainties of AAE$_{BC}$ (±0.3).

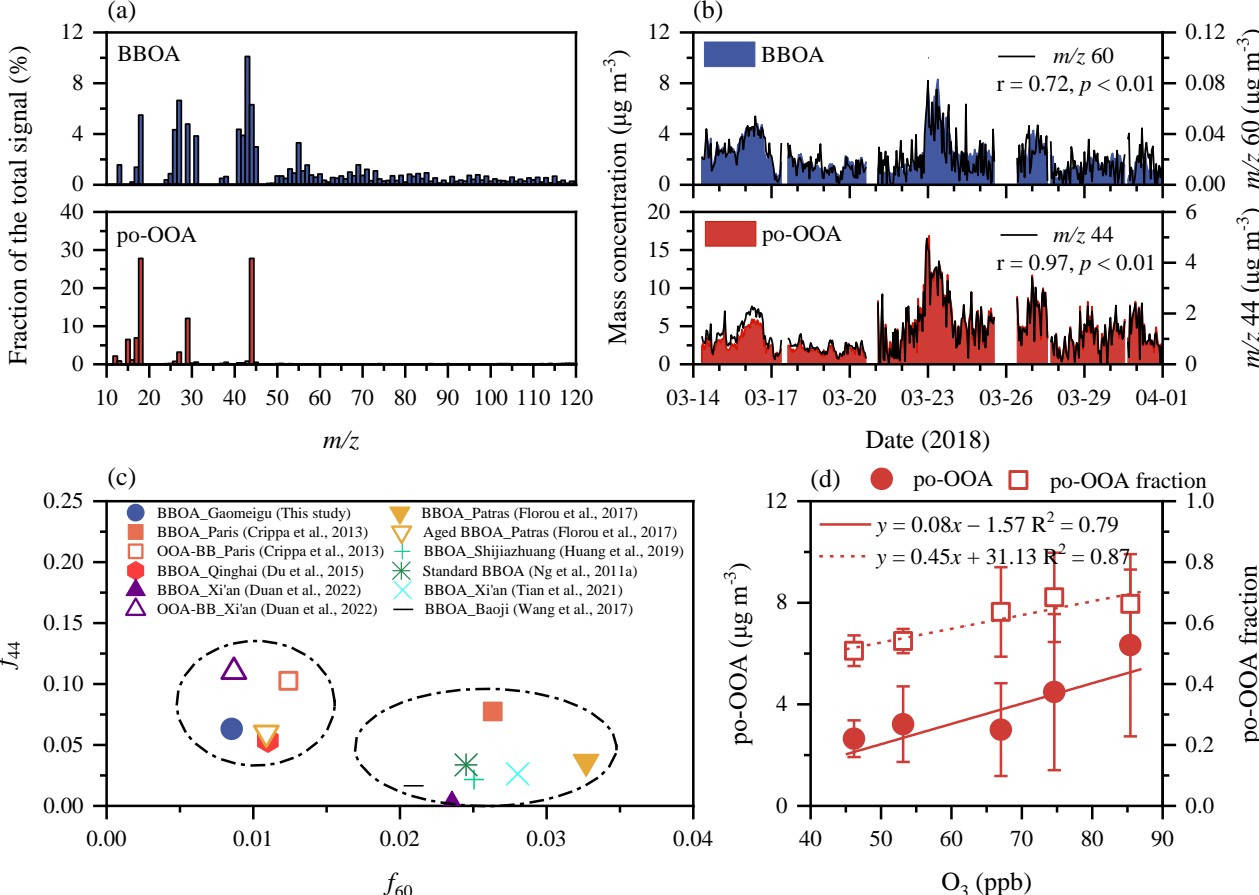

**Figure 3.** (a) Mass spectra of BBOA and po-OOA. (b) Pearson correlations between mass concentrations of OA
components and the tracer molecular fragments. (c) Scatterplots of $f_{44}$ vs. $f_{60}$ for BBOA resolved in this study and reported
by previous literatures. (d) Variations of po-OOA mass concentration and its fraction in OA as a function of $O_3$. The
data are grouped in $O_3$ bins (10 ppb increment).





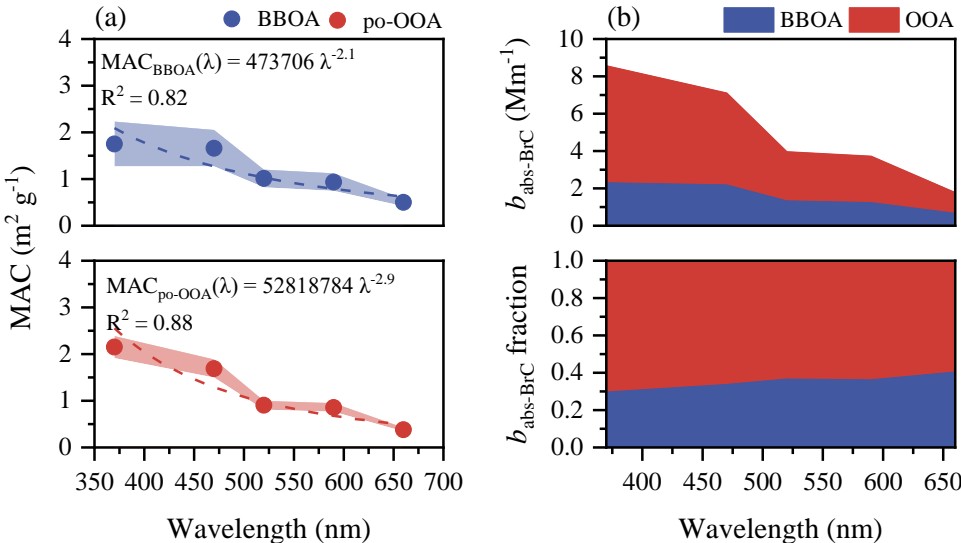

**Figure 4.** (a) The mass absorption cross section of BBOA and po-OOA ($MAC_{BBOA}$ and $MAC_{po\text{-}OOA}$, respectively) at five
wavelengths ($\lambda$ = 370, 470, 520, 590, and 660 nm). The circle and shaded area represent the mean MAC values and the
standard deviations, respectively. The dashed line is power-law fit. (b) Light absorption coefficient of BrC ($b_{abs\text{-}BrC}$) from
BBOA and po-OOA and its fraction in the total reconstructed BrC absorption at different wavelengths.





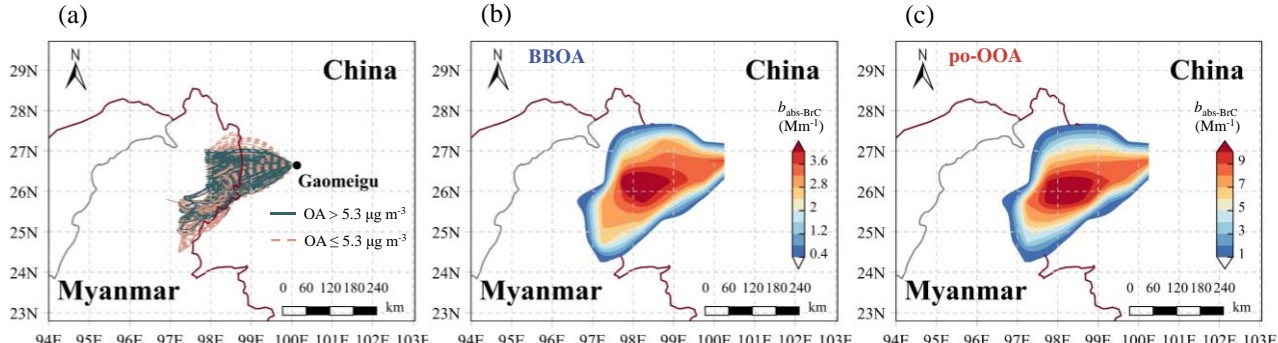

**Figure 5.** (a) 72-h backward trajectories of Gaomeigu from 8:00 on 14 to 23:00 on 31 March, 2018. (b) and (c)
Concentration weighted trajectory (CWT) maps of $b_{abs\text{-}BrC}$ at 370 nm (Mm⁻¹) from BBOA and po-OOA, respectively.

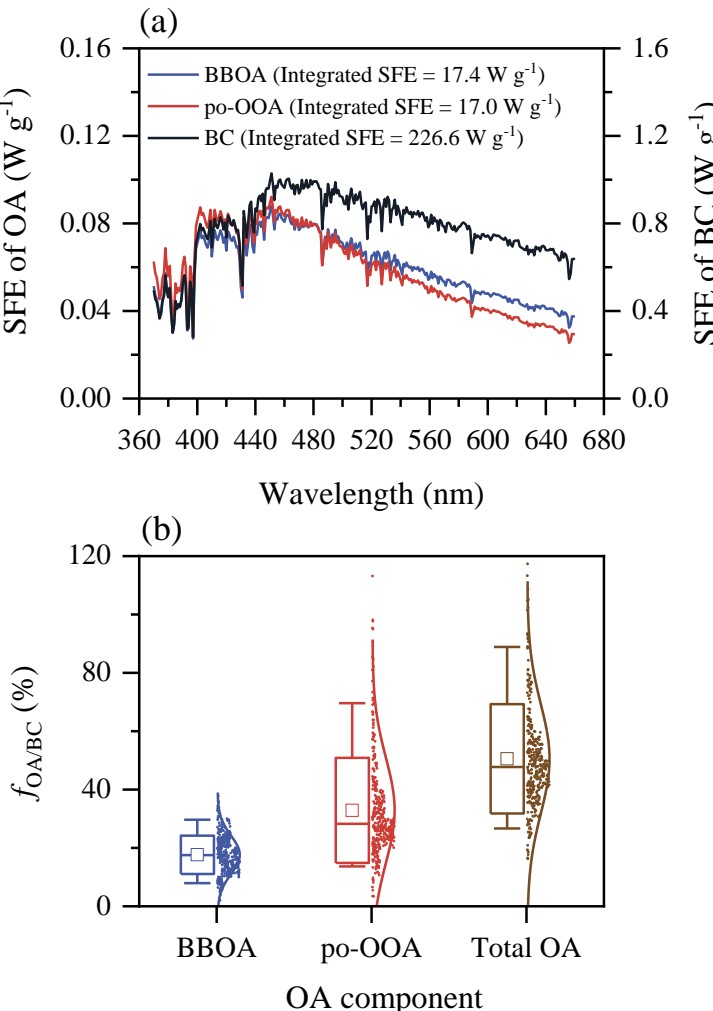

**Figure 6.** (a) Simple forcing efficiency (SFE) of light-absorbing particles from 370 to 660 nm and the integrated SFE over the entire solar spectra (370−660 nm). (b) The fraction of solar radiation absorbed by OA components relative to BC. In each panel, the short line and square inside the boxes indicate the median and mean values, respectively. The lower and upper edges of the boxes denote the standard deviation. The vertical bars ("whiskers") show the $5^{th}$ and $95^{th}$ percentiles. Scattered data points and normal distribution curve are also shown.