# Peer review of "Impacts of biomass burning and photochemical processing on the light absorption of brown carbon in the southeastern Tibetan Plateau"

_Atmospheric Chemistry and Physics, 2022_

## Author Comment (AC1)

**Responses to Referee #1:**

This paper reports on measurements of spectrally resolved particle light absorption combined with aerosol chemical composition measured at the southeastern edge of the Tibetan Plateau. The particle light absorption data are used to infer BrC and BC levels. Based on the chemical signatures the aerosol was divided into two groups and the BrC optical properties were determined for each group. Overall radiative effects of the BrC relative to BC on a per particle mass basis showed that BrC has an important role. Back trajectory analysis was used to identify source regions of the BrC. The paper is well suited for publication in this journal. The authors consider uncertainty in their approach to determine BrC, which is a nice feature of this paper. A number of suggestions for further analysis and clarifications are noted below.

**Response:** We thank the reviewer for the helpful comments and providing us the opportunity to strengthen our research. We have carefully addressed the comments in point-by-point form as shown below. Detailed responses to each of the reviewer's comments are provided in blue, and the revised text is underlined. Attached please also find the marked-up manuscript to track the changes in the revised manuscript.

**General Comments:**

**Comment (1):** Some of the methods description may be too detailed, such as the HERM algorithm description. A plainer language description would be better. Eg, line 128 and on describing Eq (3). First, this is a classic inversion problem encountered for many instruments in aerosol science (and many other fields). My understanding is that X is the measured ACSM mass spectra of the organic species, (why call it the receptor site?), G is the source contributions, which is what is you are trying to determine, and F is the mass spectra of the specific sources. Normally one knows F and performs the inversion to solve for G. In this case F is unknown, so a modified approach is used. This can then be described.

**Response:** (a) Yes, as the reviewer points out, X is the measured ACSM mass spectra of the organic species. In the original manuscript, the receptor site refers to the sampling site. To make HERM algorithm more concise and clear, we have revised the relevant description of HERM following the reviewer's suggestion:

> "HERM is an effective receptor model, which was performed to retrieve potential sources of OA in this study. The HERM algorithm groups the matrix X (measured mass spectra of organic fragments) into two nonnegative constant matrices G (source contribution) and F (mass spectra of specific sources), and the model residual matrix E, defined as:
>
> $$X = G \times F + E \qquad (3)$$
>
> The model does not require prior mass spectra of sources, and the values of G and F can be obtained using an iterative conjugate gradient algorithm. The principle of HERM has been described in detail elsewhere (Chen and Cao, 2018)." *(Page 5 Line 128–137)*

(b) We moved the Section 2.3.4 of statistical metrics to the supplement as Text S1, and also simplified the description of calculation of optical parameters (Section 2.3.3) as follows:

"The MAC of OA component in this study was resolved by the multiple linear regression (MLR) model combined with $b_{abs\text{-}BrC}$ and OA source apportionment results, which is defined as follows:

$$b_{abs\text{-}BrC}\ (\lambda) = a_1\ (\lambda) \times [\text{BBOA}] + a_2\ (\lambda)\ \times\ [\text{po-OOA}]\ (4)$$

Here, $a_1$ and $a_2$ denote the MAC of BBOA ($MAC_{BBOA}$) and po-OOA ($MAC_{po\text{-}OOA}$), respectively, in square meters per gram ($m^2\ g^{-1}$); [BBOA] and [po-OOA] are the mass concentration of BBOA and po-OOA, respectively, in micrograms per cubic meter ($\mu g\ m^{-3}$). Tolerance (0.2) and variance inflation factor (4.7) for the ordinary least square fitting results indicated that there was no serious multicollinearity between two independent variables, however, heteroscedasticity existed according to "White Test" ($p < 0.05$). Thus, the weighted least squares method was used for parameter estimation in MLR model. The MAC of BC ($MAC_{BC}$) was directly calculated with $b_{abs\text{-}BC}$ divided by the mass concentration of BC, which was obtained by dividing $b_{abs}$ (880 nm) by the default MAC (880 nm) used in the Model AE33 (Drinovec et al., 2015).

AAE describes the spectral dependence of light absorption by aerosols, and it can be calculated using a power law function with $b_{abs}$ and MAC, respectively:

$$b_{abs}\ (\lambda) = k_1 \times \lambda^{-AAE} \qquad (5)$$

$$MAC\ (\lambda) = k_2 \times \lambda^{-AAE} \qquad (6)$$

Here, $k_1$ and $k_2$ are constants independent of wavelength." *(Page 6 Line 157–Page 7 Line 178)*

**Comment (2):** A major possible issue is the characterization of the two sources identified by the source apportionment discussed above. Despite the complicated inversion, it seems the separation of source comes dow to two things, the po-OOA source characterized by *m/z* 44 (which seems similar to MO-OOA in other studies using this instrument, see discussion below), and *m/z* 60 the fragments of levoglucosan and other carbohydrates known to be emitted primarily from biomass combustion (and cooking, but which is not discussed here). The issue is one way to interpret these source apportionment results is that both are from biomass burning, BBOA is the possibly fresher or less photochemically processed BBOA and po-OOA are more aged and chemically processed BBOA. To me, this clarifies the data interpretation and is supported by the idea that both have the same source region and that they are correlated (line 262, r = 0.63). However, arguing against this is that the MACs are higher for po-

OOA which one would not be expect if this was more aged and possibly more photochemically bleached relative to the fresher or less processed (less bleached) BBOA.

**Response:** In our study, po-OOA characterized by higher $m/z$ 44 was more oxygenated than BBOA, which meant stronger photobleaching effect. We agree with the reviewer that more photobleaching would result in the lower MAC of OA (Lee et al., 2014). According to another reviewer's suggestion, we recheck the accuracy of parameter estimation of the multiple linear regression (MLR) model. We find that the heteroscedasticity in MLR model existed when we use the ordinary least square fitting previously, and this may cause the overestimation of $MAC_{po-OOA}$.

To address this concern, we use the weighted least squares method to estimate parameter of MLR model. Figure 5 shows the latest results of $MAC_{BBOA}$ and $MAC_{po-OOA}$. That is, the $MAC_{BBOA}$ is higher than the $MAC_{po-OOA}$ at wavelengths from 370 to 660 nm. In the revised manuscript, we first add the testing results and fitting method of MLR to the Sect. 2.3.3:

"Tolerance (0.2) and variance inflation factor (4.7) for the ordinary least square fitting results indicated that there was no serious multicollinearity between two independent variables, however, heteroscedasticity existed according to "White Test" ($p < 0.05$). Thus, the weighted least squares method was used for parameter estimation in MLR model." *(Page 6 Line 163–166)*

We also update the MAC results and give the explanation that po-OOA has the lower MAC compared to BBOA in the Sect. 3.3:

"Compared with BBOA, more oxygenated po-OOA was possibly more photochemically bleached, which resulted in the lower MAC (Lee et al., 2014)." *(Page 11 Line 296–297)*

In the revised manuscript, Figure 5 shows:

[Figure]

"**Figure 5.** (a) The mass absorption cross section of BBOA and po-OOA ($MAC_{BBOA}$ and $MAC_{po-OOA}$, respectively) at five wavelengths

($\lambda$ = 370, 470, 520, 590, and 660 nm). The circle and shaded area represent the mean MAC values and the standard deviations, respectively. The dashed line is power-law fit. (b) Light absorption coefficient of BrC ($b_{abs-BrC}$) from BBOA and po-OOA and its fraction in the total reconstructed BrC absorption at different wavelengths."

In the reference list, we add the new reference:

"Lee, H. J., Aiona, P. K., Laskin, A., Laskin, J., and Nizkorodov, S. A.: Effect of solar radiation on the optical properties and molecular composition of laboratory proxies of atmospheric brown carbon, Environ. Sci. Technol., 48(17), 10217−10226, https://doi.org/10.1021/es502515r, 2014."

**Specific Comments:**

**Comment (3):** When discussing Eq (1), might want to say something about how well a power law fits data.

**Response:** AAE can be calculated using a pair of light absorption coefficients ($b_{abs}$) at two different wavelength:

$$AAE(\lambda_1 \lambda_2) = \frac{\ln\left(\frac{b_{abs}(\lambda_1)}{b_{abs}(\lambda_2)}\right)}{\ln\left(\frac{\lambda_1}{\lambda_2}\right)}$$

Eq (1) is derived from this method when $b_{abs}$ at 880 nm and AAE of BC are known. Therefore, $b_{abs}$ of BC obtained from Eq (1) always satisfies the power law relationship with wavelength.

**Comment (4):** Where does the term for one of the identified sources (po-OOA) come from. Has it been used before or is it being introduced here? Essentially, it seems to be driven by mass spectral peak at $m/z$ 44, which is carboxylic acid fragment and which in past studies of ACSM/AMS is largely indicative of more aged oxygenated organic aerosol from various sources. Why not discuss this (ie, how that peak is determined in other studies and why a different name is used here).

**Response:** As noted by the reviewer, the fraction of $m/z$ 44 in OA mass spectrum ($f_{44}$) is a surrogate of oxidation degree (Aiken et al., 2008). The photochemical-oxidation processed oxygenated OA (po-OOA) resolved in this study was characterized by the highest peak at $m/z$ 44 ($f_{44}$, 27.8 %), which was quite similar to those of more-oxidized oxygenated OA (MO-OOA) ($f_{44}$ > 20%) identified frequently in previous ACSM and AMS studies (Tobler et al., 2021; Xu et al., 2018; Zhang et al., 2019). Here, since high $O_3$ was the driving factor of po-OOA formation, the term of po-OOA was introduced in this manuscript to stress the importance of photochemical processing in the Tibetan Plateau.

In the revised manuscript, to make it clear, we add:

"Another OA source was featured by the strong correlation with $m/z$ 44 (r = 0.97, $p < 0.01$), which is a surrogate of oxidation degree (Aiken et al., 2008). The most abundant peak in mass spectrum of po-OOA was at $m/z$ 44 ($f_{44}$, 27.8%), similar to those in mass spectra of more-oxidized oxygenated OA (MO-OOA) ($f_{44} > 20$ %) identified frequently in previous studies (Tobler et al., 2021; Xu et al., 2018; Zhang et al., 2019). It indicated that this OA source was likely related to extensive secondary processes occurring during transport (Wang et al., 2017; Xu et al., 2017). Figure 4d shows that both po-OOA mass concentration and its fraction in OA increased with increasing $O_3$ ($R^2$ = 0.79−0.87), however, neither of them correlated with RH (Fig. S3). These results supported that high $O_3$ was the possible driving factor of po-OOA formation, thus the term of po-OOA was introduced in this study to stress the importance of photochemical-oxidation process in the TP." *(Page 10 Line 270–279)*

In the reference list, we add the new reference:

"Aiken, A. C., DeCarlo, P. F., Kroll, J. H., Worsnop, D. R., Huffman, J. A., Docherty, K. S., Ulbrich, I. M., Mohr, C., Kimmel, J. R., Sueper, D., Sun, Y. L., Zhang, Q., Trimborn, A., Northway, M., Ziemann, P. J., Canagaratna, M. R., Onasch, T. B., Alfarra, M. R., Prévôt, A. S. H., Dommen, J., Duplissy, J., Metzger, A., Baltensperger, U., and Jimenez, J. L.: O/C and OM/OC ratios of primary, secondary, and ambient organic aerosols with High-Resolution Time-of-Flight Aerosol Mass Spectrometry, Environ. Sci. Technol., 42 (12), 4478−4485, https://doi.org/10.1021/es703009q, 2008."

**Comment (5):** In the back trajectory analysis, one could test the sensitivity to the assumed starting height of 500 m by varying this parameter over some range and see if the predicted trajectories change much?

**Response:** We do some sensitivity tests for the starting height. Figure R1 and R2 show the 72-h backward trajectories and potential source regions for $b_{abs-BrC}$ based on the assumed starting height of 750 m and 1000 m, respectively. Although there are some difference in the 72-h backward trajectories under the assumption of height of 500 m, 750, and 1000 m, the spatial distributions of potential source for $b_{abs-BrC}$ from BBOA and po-OOA were similar. Therefore, the conclusion of "The source regions with the highest CWT values were located in the northern Myanmar and along the China-Myanmar border, while the CWT values in the areas surrounding Gaomeigu were relatively low." in this study is robust and reliable.

[Figure]

**Figure R1.** (a) 72-h backward trajectories of Gaomeigu from 8:00 on 14 to 23:00 on 31 March, 2018. (b) and (c) Concentration weighted trajectory (CWT) maps of $b_{abs\text{-}BrC}$ at 370 nm (Mm$^{-1}$) from BBOA and po-OOA, respectively, based on the assumed starting height of 750 m.

[Figure]

**Figure R2.** (a) 72-h backward trajectories of Gaomeigu from 8:00 on 14 to 23:00 on 31 March, 2018. (b) and (c) Concentration weighted trajectory (CWT) maps of $b_{abs\text{-}BrC}$ at 370 nm (Mm$^{-1}$) from BBOA and po-OOA, respectively, based on the assumed starting height of 1000 m.

**Comment (6):** In the calculation of SFE, one might also do a sensitivity test on the assumed variables, possibly most importantly the albedo (also what type of ground cover does a surface albedo of 0.9 represent)?

**Response:** We are sorry that this is a writing error, and the surface albedo used in our study was 0.19. We have corrected this mistake:

"$\tau_{atm}$, F$_c$, and $a_s$ are the atmospheric transmission (0.79), the cloud fraction (0.6), and the surface albedo (0.19), respectively, which are constants from the global average calculations" *(Page 8 Line 218–219)*

For SFE calculation, the relative uncertainty of SFE can be estimated as follows:

$$U_{SFE} = \sqrt{\left(2 \times U_{\tau_{atm}}\right)^2 + U_{(1-F_c)}^2 + U_{a_s}^2}$$

Here, $U_{\tau_{atm}}$, $U_{(1-F_c)}$, and $U_{a_s}$ represent the relative uncertainties of $\tau_{atm}$, F$_c$, and $a_s$, respectively. Take an example, when $\tau_{atm}$, F$_c$, and $a_s$ have relative uncertainties of 10%,

$U_{SFE} = \sqrt{(2 \times 0.1)^2 + \left(\frac{0.6 \times 0.1}{(1 - 0.6)}\right)^2 + 0.1^2} \approx 0.27$. In this study, the atmospheric transmission, cloud fraction, and surface albedo are not measured values, we used the global average values in order to compare with other studies better.

**Comment (7):** Line 217, 18-41 fold is relative to what?

**Response:** 18−41 folds is the result of the maximum hourly $b_{abs}$ value relative to the minimum hourly $b_{abs}$ value during the sampling period. To make this point clear, we have made the following revisions:

> "The hourly $b_{abs}$ values at different wavelengths varied from minimum to maximum values by factors of 19−41 from 14 to 31 March 2018, reflecting that atmospheric environment at Gaomeigu is influenced by dynamic changes in emission sources and meteorological condition." *(Page 9 Line 228–230)*

**Comment (8):** What are the bracketed variables in Table 1?

**Response:** The bracketed variables in Table 1 represented the minimum and maximum values of hourly $b_{abs}$ during the sampling period. To make it clear, we have modified the relevant note in Table 1:

"**Table 1.** Submicron aerosol light absorption coefficient ($b_{abs}$) contributed by BrC ($b_{abs-BrC}$) and BC ($b_{abs-BC}$) at Gaomeigu during the sampling period (March 14th to 31th, 2018).

| Parameter* (Mm⁻¹) | Wavelength | | | | | |
|---|---|---|---|---|---|---|
| | 370 nm | 470 nm | 520 nm | 590 nm | 660 nm | 880 nm |
| $b_{abs}$ | $33.1 \pm 24.4$ $(4.7{-}160.0)^{**}$ | $26.7 \pm 19.7$ $(3.8{-}138.4)$ | $20.3 \pm 13.9$ $(2.6{-}93.0)$ | $18.2 \pm 12.5$ $(2.1{-}84.8)$ | $13.7 \pm 9.0$ $(1.7{-}56.9)$ | $8.0 \pm 4.9$ $(1.5{-}28.6)$ |
| $b_{abs-BrC}$ | $12.3 \pm 13.8$ | $10.7 \pm 11.5$ | $6.0 \pm 6.0$ | $5.8 \pm 5.8$ | $2.7 \pm 2.6$ | $0.0 \pm 0.0$ |
| $b_{abs-BC}$ | $20.8 \pm 12.8$ | $16.0 \pm 9.8$ | $14.3 \pm 8.8$ | $12.4 \pm 7.7$ | $11.0 \pm 6.8$ | $8.0 \pm 4.9$ |

*$b_{abs-BrC}$ and $b_{abs-BC}$ were calculated based on the $AAE_{BC} = 1.1$.
**The measured hourly $b_{abs}$ from minimum to maximum values."

**Comment (9):** The AAE frequency distribution is stated to be normally distributed (line 225), but it looks possibly bimodal. What is the justification for stating it is normally distributed?

**Response:** Thank you for pointing this out. We did the Kolmogorov-Smirnov and Shapiro-Wilk tests for AAE values with $p$ values less than 0.01. The results proved that the frequency distribution of AAE was not normally distributed. The relevant description has been revised as follows:

> "Frequency histograms of hourly AAE values showed the media AAE value of 1.59 with interquartile range from 1.38 to 1.83 (Fig. S1). Over 72 % of AAE values were higher than 1.4 (Upper limit of $AAE_{BC}$), implying the presence of both BrC and BC absorption in the submicron aerosol at Gaomeigu." *(Page 9 Line 237–238)*

**Comment (10):** Line 259 and on discussing po-OOA correlation with $O_3$ and RH. The logic here is not clear. Both $O_3$ and RH vary substantially both spatially and even diurnally. Given this, how can $O_3$ and RH at the measurement site be used to infer what the particles were exposed to over the time when transported from source region to where measured? It is the history of what the particles were exposed to that determines the properties at the measurements, the conditions at the measurement site may have only a small or minor impact.

**Response:** Yes. In fact, we are unable to elucidate what reactions occur in the secondary OA during atmospheric processes with the data measured in the sampling site alone. Establishing the relationship between indicators of oxidation (e.g., $O_3$, $O_x$, RH, and aerosol liquid water content) and secondary OA is the useful way to explore the potential formation mechanisms, but it is still not 100% confirmed whether these indicators are involved in secondary formation. In this study, both po-OOA mass concentration and its fraction in OA increased with increasing $O_3$ ($R^2 = 0.79{-}0.87$), however, neither of them correlated with RH. These results indicated, to some extent, that high $O_3$ was the possible driving factor of po-OOA formation. Meanwhile, the intense photochemical environment is an inherent feature of the TP. Thus, we believe that photochemical oxidation is an important potential pathway for the formation of po-OOA.

**Comment (11):** Line 261 to 264, seems that another explanation is that most of the po-OOA is processed BBOA, as discussed above.

**Response:** As addressed in the comment (2), in the revised manuscript, po-OOA was more oxygenated than BBOA, and had the lower MAC possibly due to more photobleaching effect.

**Comment (12):** The last statement of the Discussion (line 319) and in the conclusions is not clear. That is, how does knowledge of the secondary BrC help tackling climate change? Be more specific, this is too general a statement to be meaningful.

**Response:** Thank you for this comment. The revised discussion shows:

[revised manuscript text omitted]

---

## Author Comment (AC2)

**Responses to Referee #2:**

**General comments:**

This manuscript presents a comprehensive analysis of light-absorbing aerosols, especially organic aerosol, in the southeastern margin of Tibetan Plateau during the pre-monsoon season. Authors found light absorption of BrC was mainly contributed by biomass burning and secondary formation. Furthermore, radiative effect of BrC was found to be non-negligible compared to that of BC. Generally, in view of the concern about the melting of plateau glaciers caused by global warming, it is of great significance to evaluate the optical properties, sources and formation processes of BrC in the TP. The results of this study provide valuable dataset and analysis to illustrate the importance of BrC on climate change in the TP, which will be helpful in model simulation of radiative effect of aerosols. Overall, this manuscript is well written and results are interesting with novelty. I recommend this manuscript should be accepted for publication after addressing the following issues.

**Response:** We highly appreciate the thoughtful and valuable suggestions by the reviewer, which are helpful for us to improve the quality of our manuscript. We have carefully addressed the comments in point-by-point form as shown below. Detailed responses to each of the reviewer's comments are provided in blue, and the revised text is underlined. Attached please also find the marked-up manuscript to track the changes in the revised manuscript.

**Major comments:**

**Comment (1):** In this study, the mass spectra of po-OOA were characterized by high *m/z* 44 and low *m/z* 43. This indicated po-OOA was more-oxidized, which was similar to MO-OOA identified in AMS and ACSM studies. Generally, MO-OOA has lower MAC compared to primary OA due to photo-bleaching effect; however, in this study the estimated MAC of po-OOA was larger than that of BBOA. Could the authors explain it? On the other hand, I suggest authors could conduct the tests of collinearity and heteroscedasticity of independent variables considering the good correlation between BBOA and po-OOA, and this would guarantee the accuracy of multiple linear regression as shown in Eq. 5.

**Response:** We agree with the reviewer that the more-oxidized OOA might have the lower MAC compared to primary OA due to photo-bleaching effect. We thus have conducted multicollinearity and heteroscedasticity tests for the multiple linear regression (MLR) model, following the reviewer's suggestion. We find that there is no serious multicollinearity between two independent variables ([BBOA] and [po-OOA]), however, the heteroscedasticity existed when using the ordinary least square fitting.

To address this concern, we use the weighted least squares method to estimate parameter of MLR model. Figure 5 shows the latest results of $MAC_{BBOA}$ and $MAC_{po-OOA}$. That is, the $MAC_{BBOA}$ is higher than the $MAC_{po-OOA}$ at wavelengths from 370 to 660 nm. The fact that po-OOA is more oxygenated than BBOA means more photobleaching on po-OOA. This might result in the lower MAC of po-OOA (Lee et

al., 2014). In the revised manuscript, we first add the testing results and fitting method of MLR to the Sect. 2.3.3:

> "Tolerance (0.2) and variance inflation factor (4.7) for the ordinary least square fitting results indicated that there was no serious multicollinearity between two independent variables, however, heteroscedasticity existed according to "White Test" ($p < 0.05$). Thus, the weighted least squares method was used for parameter estimation in MLR model." *(Page 6 Line 163–166)*

We also update the MAC results and give the explanation that po-OOA has the lower MAC compared to BBOA in the Sect. 3.3:

> "Compared with BBOA, more oxygenated po-OOA was possibly more photochemically bleached, which resulted in the lower MAC (Lee et al., 2014)." *(Page 11 Line 296–297)*

In the revised manuscript, Figure 5 shows:

[Figure]

> "**Figure 5.** (a) The mass absorption cross section of BBOA and po-OOA ($MAC_{BBOA}$ and $MAC_{po-OOA}$, respectively) at five wavelengths ($\lambda$ = 370, 470, 520, 590, and 660 nm). The circle and shaded area represent the mean MAC values and the standard deviations, respectively. The dashed line is power-law fit. (b) Light absorption coefficient of BrC ($b_{abs-BrC}$) from BBOA and po-OOA and its fraction in the total reconstructed BrC absorption at different wavelengths."

In the reference list, we add the new reference:

> "Lee, H. J., Aiona, P. K., Laskin, A., Laskin, J., and Nizkorodov, S. A.: Effect of solar radiation on the optical properties and molecular composition of laboratory proxies of atmospheric brown carbon, Environ. Sci. Technol., 48(17), 10217−10226, https://doi.org/10.1021/es502515r, 2014."

**Comment (2):** The amount of solar energy absorbed by BrC relative to BC ($f_{BrC/BC}$) was used to evaluate the radiative effect of BrC. In this study, the $f_{BrC/BC}$ was calculated by "simple forcing efficiency" and mass concentration. In fact, there are many studies have been conducted by $f_{BrC/BC}$ calculation according to the method reported by Kirillova et al. (2014, JGR-A). Have the authors compared these two approaches? Furthermore, the uncertainties of $f_{BrC/BC}$ need to be discussed. The uncertainties can be caused by e.g., the errors of assumption parameter, MAC and mass concentration.

**Response:** Following the Kirillova et al. (2014), the fraction of solar energy absorbed by BrC relative to BC ($f_{BrC/BC}$) can be calculated as follows:

$$f_{BrC/BC} = \frac{\int_{370}^{660} S_0(\lambda)\{1 - e^{-(MAC_{BrC}(\lambda)) \times C_{BrC} \times h_{ABL}}\}d\lambda}{\int_{370}^{660} S_0(\lambda)\{1 - e^{-(MAC_{BC}(\lambda)) \times C_{BC} \times h_{ABL}}\}d\lambda}$$

Here, $S_0$ is the solar irradiance based on the ASTM G173-03 reference spectra; MAC is the mass absorption cross section; $C_{BrC}$ and $C_{BC}$ are the mass concentrations of BrC and BC; $h_{ABL}$ is the height of the atmospheric boundary layer, which is set to 1000 m. Based on the Kirillova's method, the average fractional radiative forcing by two OAs relative to BC was 45.0 %, which is close to the result (48.8 %) calculated by "simple forcing efficiency" and mass concentration in the manuscript. This suggests that both methods are reasonable and valid in assessing the radiative effect of BrC relative to BC.

$f_{BrC/BC}$ is a relative value, and its uncertainties come from the errors of MAC and mass concentration. Considering that MAC with a 1 nm resolution is extrapolated using the power law fitting in this study, we gave the uncertainties in $f_{BrC/BC}$ caused by the variations in mass concentration as shown in Figure 7.

In the revised manuscript, Figure 7 shows:

[Figure]

"**Figure 7.** (a) Simple forcing efficiency (SFE) of light-absorbing particles from 370 to 660 nm and the integrated SFE over the entire

solar spectra (370−660 nm). (b) The fraction of solar radiation absorbed by OA components relative to BC. In each panel, the short line and square inside the boxes indicate the median and mean values, respectively. The lower and upper edges of the boxes denote the standard deviation. The vertical bars ("whiskers") show the 5$^{th}$ and 95$^{th}$ percentiles. Scattered data points and normal distribution curve are also shown."

**Comment (3):** From the results of this study, biomass burning emission and photochemical reactions both contributed to the formation of BrC. Since the intense photochemical environment is an inherent feature of the TP, it is difficult to control the photochemical production of BrC. Thus, effective control measures are to reduce the anthropogenic emissions of biomass burning. The authors should add some discussions at the end of the results to imply the significance of this work.

**Response:** Thank you for point this out. We add the relevant description in the revised discussion:

"The fractional radiative forcing by two OAs relative to BC was as high as 48.8 ± 15.5 %, in which the relative radiative forcing of po-OOA to BC (24.2 ± 13.2 %) was almost equal that of BBOA to BC (24.6 ± 9.1 %) (Fig. 7b). These results suggested that BrC emitted from biomass burning and formed by photochemical oxidation was an efficient radiative forcing agent, which, along with BC, can remarkably disturb the radiative balance over the TP. Thus, the inclusion of BrC in the climate models will provide a better understanding of climate change of the southeastern TP. It should also be noted that although BBOA and po-OOA had similar radiative effects, effective measures on tackling the impact of BrC are to reduce primary emission and volatile organic precursor of BrC from biomass burning in the future, since the intense photochemical environment is an inherent feature of the TP." *(Page 12 Line 340–Page 13 Line 348)*

**Specific Comments:**

**Comment (4):** Line 12: Replace "as well as" with "compared to".

**Response:** We have replaced "as well as" with "compared to" as follows:

"Brown carbon (BrC) in the atmosphere can greatly influence aerosol's radiative forcing over the Tibetan Plateau (TP), because it has the non-negligible capacity of light absorption compared to black carbon (BC);" *(Page 1 Line 11–12)*

**Comment (5):** Line 25: Replace "BrC from biomass burning emissions" with "BrC of biomass burning origin".

**Response:** We have replaced "BrC from biomass burning emissions" with "BrC of biomass burning origin" as follows:

"This study highlighted a significant influence of BrC of biomass burning origin and secondary formation on climate change over the TP region during the pre-monsoon season." *(Page 1 Line 26–28)*

**Comment (6):** Line 38-39: Add citation to the statement of "In particular, the atmospheric heating of BrC is greater than that of BC in the tropical mid and upper troposphere.".

**Response:** The statement of "In particular, the atmospheric heating of BrC is greater than that of BC in the tropical mid and upper troposphere." is also cited from Zhang et al. (2020a). To make it clear, we revised the text as follows:

"Zhang et al. (2020a) estimated that globally BrC contributed more than 25% of BC RF, wherein the atmospheric heating of BrC is greater than that of BC in the tropical mid and upper troposphere." *(Page 2 Line 38–40)*

**Comment (7):** Line 66-67: Replace "plateau" with "TP". Full name of TP is already mentioned in Line 56.

**Response:** We have replaced "plateau" with "TP". It now reads:

"For example, Zhang et al. (2015) reported that biomass burning from South Asia has the largest impact on BC in the central TP, while fossil fuel combustion contributed the most to BC burden in the northeast TP in all seasons and southeast TP in the summer." *(Page 3 Line 66–68)*

**Comment (8):** Line 182: Check the publication year of Draxler and Hess.

**Response:** We have corrected this citation information.

"The trajectories were calculated with the Hybrid Single-Particle Lagrangian Integrated Trajectory (HYSPLT) model developed by the National Oceanic and Atmospheric Administration (Draxler and Hess, 1998)." *(Page 7 Line 192–194)*

**Comment (9):** Line 216: Please consider to put Fig. Sl in the main figures of manuscript. Fig. S1 indicates the periods of high concentrations and $b_{abs}$ that later discussed in this manuscript.

**Response:** We have moved Fig. S1 to the main manuscript and assigned as Fig. 2.

**Comment (10):** Line 256: Replace "as well as" with "similar to".

**Response:** We have replaced "as well as" with "similar to" as follows:

"The most abundant peak in mass spectrum of po-OOA was at $m/z$ 44 ($f_{44}$, 27.8 %), similar to those in mass spectra of more-oxidized oxygenated OA (MO-OOA) ($f_{44} > 20$ %) identified frequently in

previous studies (Tobler et al., 2021; Xu et al., 2018; Zhang et al., 2019)." *(Page 10 Line 272–274)*

**Comment (11):** Line 261: Replace "affect most" with "strongly affect".

**Response:** We have deleted this sentence.

**Comment (12):** Line 283: How the $rb_{abs-BrC}$ was calculated? Is it reconstructed from the $b_{abs}$ of BBOA and po-OOA? Have the authors compared the $rb_{abs-BrC}$ with the calculated $b_{abs-BrC}$ from Eq. 2?

**Response:** Yes, $rb_{abs-BrC}$ is calculated by the sum of the reconstructed $b_{abs}$ from BBOA and po-OOA based on the MLR model. To express this point clear, the following sentence has been added in the revised manuscript:

[revised manuscript text omitted]

---

## Author Response (AR2)

**Responses to the Editor:**

Dear authors,

Thank you for addressing the Referee's comments. There are two related comments from Referee #1 I would like you to address more explicitly. Specifically, I would like you to more explicitly address Comments 2 and 11 from Referee #1. I believe the Referee is suggesting that the component you call "BBOA" is really "unaged BBOA" and the component you call "po-OOA" is really "aged BBOA". Please discuss in the revised manuscript if this is a possible explanation of your results and why or why not. Thank you for considering.

Sincerely,

Allan Bertram

**Response:** Thank you for the helpful comment and providing us the opportunity to revise the manuscript. Detailed responses to each comment are provided in blue, and the revised text is underlined. Attached please also find the marked-up manuscript to track the changes in the revised manuscript.

BBOA was identified by the prominent peak of $m/z$ 60, which is good organic tracer of biomass burning. In this study, BBOA was less aged by the analysis of the fraction of $m/z$ 44 vs. $m/z$ 60 ($f_{44}$ vs. $f_{60}$) in BBOA mass spectrum as follows:

> "Scatterplots of $f_{44}$ vs. $f_{60}$ was used to analyze aging degree of BBOA in the atmosphere (Fig. 4c). The $f_{60}$ usually decreases from fresh to aged biomass burning emissions because of degradation and oxidation reactions during the atmospheric aging, while the $f_{44}$ increases (Paglione et al., 2020). The $f_{60}$ and $f_{44}$ of BBOA resolved in this study (0.9 % and 6.3 %, respectively) indicates BBOA was less aged, possibly caused by the long-distance regional transport." *(Page 10 Line 265–269)*

This is consistent with the view that BBOA is the possibly less chemically processed as mentioned by the Referee #1 in comment (2).

Additionally, po-OOA characterized by the high peak of $m/z$ 44 was an oxygenated OA (OOA) that was resolved using a receptor model with $m/z$ fragments from 12 to 120 (method Sect. 2.3.2). Using this method, it is challenging in linking the OOA to its corresponding primary sources (Canonaco et al., 2013). In this study, BBOA and po-OOA have the similar potential source region (Fig. 6b and c) and moderate correlation between their concentrations (r = 0.63, $p < 0.01$), thus we conclude that oxidation of volatile organic precursor from biomass burning may contribute to part of po-OOA:

> "Moreover, the temporal variation of mass concentration in po-OOA moderately correlated with that in BBOA (r = 0.63, $p < 0.01$), indicating that a portion of po-OOA could be derived from oxidation of volatile organic precursor from biomass burning (Bruns et al., 2016; Posner et al., 2018)." *(Page 10 Line 279–282)*

However, due to the method limitation, we cannot over-interpret the data and conclude

that po-OOA is really "aged BBOA". If the editor and reviewer allow, we prefer to use "po-OOA" as it was.

To clarify this concern, we have re-answered comments (2) and (11) more explicitly as shown below.

**Responses to Referee #1:**

**Comment (2):** A major possible issue is the characterization of the two sources identified by the source apportionment discussed above. Despite the complicated inversion, it seems the separation of source comes dow to two things, the po-OOA source characterized by $m/z$ 44 (which seems similar to MO-OOA in other studies using this instrument, see discussion below), and $m/z$ 60 the fragments of levoglucosan and other carbohydrates known to be emitted primarily from biomass combustion (and cooking, but which is not discussed here). The issue is one way to interpret these source apportionment results is that both are from biomass burning, BBOA is the possibly fresher or less photochemically processed BBOA and po-OOA are more aged and chemically processed BBOA. To me, this clarifies the data interpretation and is supported by the idea that both have the same source region and that they are correlated (line 262, r = 0.63). However, arguing against this is that the MACs are higher for po-OOA which one would not be expect if this was more aged and possibly more photochemically bleached relative to the fresher or less processed (less bleached) BBOA.

**Response:** (1) As the reviewer points out, po-OOA in this study was characterized by the highest peak at $m/z$ 44 ($f_{44}$, 27.8 %), which was quite similar to those of more-oxidized oxygenated OA (MO-OOA) ($f_{44} > 20\%$) identified frequently in previous ACSM and AMS studies. We have add the the relevant discussion in the revised manuscript:

> "Another OA source was featured by the strong correlation with $m/z$ 44 (r = 0.97, $p < 0.01$), which is a surrogate of oxidation degree (Aiken et al., 2008). The most abundant peak in mass spectrum of po-OOA was at $m/z$ 44 ($f_{44}$, 27.8%), similar to those in mass spectra of more-oxidized oxygenated OA (MO-OOA) ($f_{44} > 20$ %) identified frequently in previous studies (Tobler et al., 2021; Xu et al., 2018; Zhang et al., 2019)." *(Page 10 Line 270–274)*

(2) The cooking OA (COA) was ubiquitously resolved at urban sites, but not common in remote and regional background sites (e.g., Tibetan Plateau in this study) (Zhou et al., 2020 and references therein). Same to previous studies, COA is not resolved in this study, probably due to the sparse population in the Tibetan Plateau and thus much reduced influences from cooking activities.

(3) As for the comment "BBOA is the possibly fresher or less photochemically processed BBOA and po-OOA are more aged and chemically processed BBOA", we have addressed this question in the response to editor. In summary, BBOA was less

aged, and po-OOA was more oxygenated, where a portion of po-OOA was associated with biomass burning.

(4) Moreover, as the reviewer expected, MACs of BBOA are larger than those of po-OOA, as we have addressed in the 1st-round revision:

In our study, po-OOA characterized by higher $m/z$ 44 was more oxygenated than BBOA, which meant stronger photobleaching effect. We agree with the reviewer that more photobleaching would result in the lower MAC of OA (Lee et al., 2014). According to another reviewer's suggestion, we recheck the accuracy of parameter estimation of the multiple linear regression (MLR) model. We find that the heteroscedasticity in MLR model existed when we use the ordinary least square fitting previously, and this may cause the overestimation of $MAC_{po\text{-}OOA}$.

To address this concern, we use the weighted least squares method to estimate parameter of MLR model. Figure 5 shows the latest results of $MAC_{BBOA}$ and $MAC_{po\text{-}OOA}$. That is, the $MAC_{BBOA}$ is higher than the $MAC_{po\text{-}OOA}$ at wavelengths from 370 to 660 nm. In the revised manuscript, we first add the testing results and fitting method of MLR to the Sect. 2.3.3:

"Tolerance (0.2) and variance inflation factor (4.7) for the ordinary least square fitting results indicated that there was no serious multicollinearity between two independent variables, however, heteroscedasticity existed according to "White Test" ($p < 0.05$). Thus, the weighted least squares method was used for parameter estimation in MLR model." *(Page 6 Line 163–166)*

We also update the MAC results and give the explanation that po-OOA has the lower MAC compared to BBOA in the Sect. 3.3:

"Compared with BBOA, more oxygenated po-OOA was possibly more photochemically bleached, which resulted in the lower MAC (Lee et al., 2014)." *(Page 11 Line 296–297)*

In the revised manuscript, Figure 5 shows:

[Figure]

"**Figure 5.** (a) The mass absorption cross section of BBOA and po-OOA ($MAC_{BBOA}$ and $MAC_{po\text{-}OOA}$, respectively) at five wavelengths ($\lambda$ = 370, 470, 520, 590, and 660 nm). The circle and shaded area represent the mean MAC values and the standard deviations, respectively. The dashed line is power-law fit. (b) Light absorption coefficient of BrC ($b_{abs\text{-}BrC}$) from BBOA and po-OOA and its fraction in the total reconstructed BrC absorption at different wavelengths."

In the reference list, we add the new reference:

"Lee, H. J., Aiona, P. K., Laskin, A., Laskin, J., and Nizkorodov, S. A.: Effect of solar radiation on the optical properties and molecular composition of laboratory proxies of atmospheric brown carbon, Environ. Sci. Technol., 48(17), 10217−10226, https://doi.org/10.1021/es502515r, 2014."

**Comment (11):** Line 261 to 264, seems that another explanation is that most of the po-OOA is processed BBOA, as discussed above.

**Response:** As the response to the editor, we cannot quantify how much of the po-OOA is processed BBOA due to the method limitation. We can only conclude that oxidation of volatile organic precursor from biomass burning may contribute to part of po-OOA:

"Moreover, the temporal variation of mass concentration in po-OOA moderately correlated with that in BBOA (r = 0.63, $p$ < 0.01), indicating that a portion of po-OOA could be derived from oxidation of volatile organic precursor from biomass burning (Bruns et al., 2016; Posner et al., 2018)." *(Page 10 Line 279–282)*

[revised manuscript text omitted]